# Indoor Human Detection from a Building’s Exterior Using 433 MHz Wireless Transceivers

**DOI:** 10.3390/s23146280

**Published:** 2023-07-10

**Authors:** Sunghoon Jo, Sehee Park, Gu-In Kwon

**Affiliations:** Department of Electrical and Computer Engineering, Inha University, Incheon 22212, Republic of Korea; csh05132003@gmail.com (S.J.); sw_parksehee@inha.edu (S.P.)

**Keywords:** wireless sensor networks, RSSI, through-the-wall, human detection

## Abstract

This study introduces a novel system for detecting humans inside a building by utilizing RF signals from the building’s exterior. Existing RF communication devices encounter signal attenuation issues when passing through walls, limiting their effectiveness. In contrast, our system employs a low-power, long-distance communication signal operating at 433 MHz to enhance signal permeability, enabling the accurate detection of individuals within the building. The system analyzes received signal strength indicator (RSSI) data using variance and mean analysis algorithms to determine the presence or absence of people. The evaluation results indicate promising average accuracies of 88% for the variance analysis algorithm and 97.7% for the mean analysis algorithm. The proposed system holds potential for real-world deployment, particularly in challenging scenarios such as fire incidents, where pre-installation is challenging. Continued research and development efforts aim to enhance the system’s performance and address any limitations, making it more effective and robust in various practical applications.

## 1. Introduction

Technology for human detection has evolved continuously since the early 20th century. Early systems relied on heat detection, light sensors, infrared sensors, and ultrasonic sensors, but their range was limited, and they were prone to malfunctions caused by factors such as wind, noise, ultraviolet rays, and nearby heat sources [1,2]. However, the emergence of vision-based systems utilizing image and video processing techniques has significantly improved human detection and identification. These systems offer wider detection ranges and the ability to recognize detailed positions [3,4]. However, vision-based systems rely on visual data, which can be influenced by environmental conditions at the camera installation site. This introduces complexity and higher costs in terms of algorithm implementation and equipment maintenance.

In recent years, there has been growing interest in utilizing wireless signals to overcome the limitations of traditional systems. This approach involves analyzing the Received Signal Strength Indicator (RSSI) of Wi-Fi or Bluetooth signals to detect changes caused by human movement. By leveraging existing wireless network infrastructure, this technology offers relatively lower costs and is less affected by environmental factors compared with previous technologies. Wireless-signal-based human detection has various potential applications, including smart home automation and indoor navigation systems [5,6]. However, it is still an emerging field, and several technical challenges need to be addressed before widespread deployment in real-world scenarios.

Radio frequency (RF)-based human detection techniques can be affected by various environmental factors, such as multipath fading caused by signal reflection, refraction, and diffraction, which can make it difficult to differentiate between human presence and environmental variations [7]. One of the key challenges with these RF-based techniques is the limited detection range, especially in situations where people are behind multiple obstacles such as walls. To overcome this limitation, researchers have explored various methods, including using multi-hop nodes or drones to expand the detection range. In the study by Dhekne et al. [8], multi-hop nodes were used to improve the detection range of RF signals and identify sensing targets deep inside the building. This method involves deploying a network of nodes that can relay signals to each other to cover a larger area and improve the detection range. Similarly, the study by Ma et al. [9] explored the use of drones to broaden the detection range. Drones can fly over the area of interest and use RF-based sensors to detect human presence. However, one of the major limitations of these methods is that they are vulnerable to node failure or changes. This can make it difficult to install and use these systems outside the building, where environmental factors can be even more complex and unpredictable. Therefore, more research is needed to develop robust and reliable RF-based human detection systems that can overcome these limitations and be deployed in a wide range of indoor and outdoor environments.

In this paper, we present a new approach for human detection using multiple low-power transceivers using low-power 433 MHz frequency bands. This approach aims to overcome the challenges posed by obstacles such as multiple walls and other building materials that can block or weaken wireless signals. Unlike existing protocols such as Wi-Fi and Bluetooth, which cannot detect humans in scenarios where signals need to pass through multiple walls [10], the 433 MHz frequency band operates at a lower frequency, making it more suitable for long-range communication and more effective at penetrating obstacles. Therefore, our final goal was to have both the transmitter and receivers positioned outside the building, as depicted in Figure 1, with the wireless signal being transmitted into the building to detect the presence of people inside. The received data are then analyzed using algorithms to determine whether a person is present within the detection range between the transmitter and receiver. Using a low-power 433 MHz frequency band can help overcome the challenges of signal attenuation and penetration through obstacles, which can improve the accuracy and range of human detection. Prior to deploying our human detection system, it was essential to conduct a setup phase to determine some optimal threshold values. This involved measuring the RSSI values in both the presence and absence of people, ensuring that the building was empty during the measurements.

## 2. Background

In the field of RF wireless communication, there have been various studies on detecting human presence using wireless signals. Multiple wireless communication protocols, such as Wi-Fi, Bluetooth Low Energy (BLE), ultra-wideband (UWB), and Long Range (LoRa), have been used to detect humans in different ways. In this section, we introduce several prior studies.

Wi-Fi is the most widely used wireless communication technology for human detection, and uses frequencies of 2.4 GHz and 5 GHz. Initially, human detection was achieved using RSSI, but later, using Channel State Information (CSI), detailed human activity information could be detected. Orthogonal Frequency Division Multiplexing (OFDM) is a method of multiplexing a transmitted data sequence into many narrowband sub-carriers, and CSI represents the frequency response characteristics of each OFDM sub-carrier channel. CSI contains information about signal attenuation, diffraction, reflection, and other signal distortions between the transmitter and receiver; therefore, it enables more flexible adaptation to temporal environmental changes, resulting in better localization performance compared with RSSI [11,12]. In the study by Sigg et al. [13], they used K-Nearest Neighbor (KNN) classifiers to differentiate various bodily activities based on Wi-Fi signal RSSI, achieving an 80% accuracy rate. Additionally, in the study by Abdelnasser et al. [14], they achieved 96% recognition accuracy using RSSI values to detect various hand gestures. In the study by Chen et al. [15], an attention-based bidirectional long short-term memory (ABLSTM) approach was used to recognize human activity using Wi-Fi CSI measurements and validate its effectiveness. This study achieved high accuracy in differentiating various bodily activities such as falling, walking, running, falling, sitting, and standing. In the study by Xiao et al. [16], they researched a CSI-based indoor positioning system, adopting backpropagation and K-mean algorithms for real-time differentiation between Line-of-Sight (LOS) and Non-Line-of-Sight (NLOS) in human detection. These studies using CSI pursue more refined detection beyond merely detecting human presence [17,18,19,20,21,22].

BLE, a low-power version of Bluetooth introduced in Bluetooth version 4.0, operates in the 2.4 GHz frequency band with 40 channels spaced 2 MHz apart. It utilizes Frequency Hopping Spread Spectrum (FHSS) to mitigate noise interference by dynamically changing channels 1600 times per second. In the study by Brockmann et al. [23], RSSI variance and mean analysis algorithms with predetermined thresholds were employed to detect the presence or absence of a person between the transmitter and receiver in real time. The study by Münch et al. [24] involved a system using multiple transmitters and receivers to detect and count people in a classroom. RSSI values were collected and classified into different datasets; logistic regression, KNN, and Support Vector Machine (SVM) models achieved over 90% accuracy in detecting people when using the normalized dataset. In S. Naghdi’s study [25], Radial Basis Function (RBF) and Multi-Layer Perceptron (MLP) networks were utilized to detect people, with MLP networks demonstrating higher accuracy. Ongoing studies [26,27,28,29] explore Bluetooth’s potential for person detection, aiming for accuracy rates of 90% or higher across various models.

UWB is a wireless technology that operates in the frequency range of 100 MHz to 3 GHz. In a study by S. D. Liang [30], two experiments were conducted using UWB radar to detect a person behind a 1-foot thick wall and a 2-inch thick wooden door. The researchers detected chest movements during breathing and enhanced the weak radar echo signal using the standard deviation (std) of the breathing signal. By comparing the calculated std value to a threshold, they determined the presence or absence of a person. Another study by Yusuf et al. [31] used UWB to estimate the number of people below a ship’s deck, achieving an 88% accuracy. In a study by Li et al. [32], UWB was used to measure the signal-to-noise ratio (SNR) for locating survivors in disaster situations, successfully detecting individuals behind a 20 cm plaster wall at distances of 1.4 m and 1.9 m.

LoRa is a wireless communication technology that typically uses a frequency range of 800–950 MHz, which varies by country. LoRa is known for its long-range connectivity and low power consumption. The WIDESEE system [33] uses a drone equipped with a single transmitter–receiver to detect the presence of people by analyzing the reflected signals of the transmitter. However, LoRa can be susceptible to interference due to its wide detection range. To address this issue, the Power Spectral Density (PSD) is utilized. The PSD is obtained by applying FFT to the received signal, which provides information about the frequency strength, frequency mixture, and noise level. After multiplying the transformed FFT results with complex numbers and computing the average, the PSD can be obtained. PSD is useful for comparing the vibration levels of signals with different data counts. By calculating the PSD at different frequencies and normalizing it, the normalized PSD value of a specific frequency can approach 1, which can be used to detect the presence of a person based on the frequency with the highest PSD. The second-order Butterworth low-pass filter is used to filter out unwanted frequencies, such as drone vibrations and interference, to ensure accurate detection. The WIDESEE system has been shown to detect people up to a maximum distance of 53 m in an open square and can detect people inside a high-rise building surrounded by a 5 cm thick window and 40 cm thick concrete walls.

There are also studies that utilize low-cost Software-Defined Radio (SDR) solutions. In the study by Uysal et al. [34], the focus is on a dual Through-the-Wall scenario where both the transmitter and receiver are placed outside the walls of the monitored area. The experiments are conducted by installing the transmitter and receiver on the outer sides of the walls of rooms measuring 2.8 × 3.8 m^2^ and 6.1 × 9 m^2^, respectively. A wideband 900 MHz single-frequency signal is used, which passes through the walls twice, and machine learning (ML) is employed to classify whether there is movement between the walls, only breathing without movement, or no presence of a person. RSSI is utilized, which is preprocessed through filtering and downsampling. To efficiently estimate the Respiratory Rate (RR) from the attenuated signals due to the double walls, a Non-Linear Least Squares (NLS) approach is employed. A machine-learning-based Decision Tree classifier is then used, considering the difference between the variance of the RR estimation and the variance in the observed data as a discriminative criterion, achieving an accuracy of over 99%. In the study by Jacob et al. [35], a system using a 3.7 GHz frequency is implemented, where the antenna and the surrounding environment are modified to reliably detect faint radiation emitted by humans. The system can detect individuals behind wooden doors at distances of up to 2 m. In the study by Taylor et al. [36], a person detection system is developed for applications such as elderly care by caregivers. Their system utilizes Universal Software Radio Peripheral (USRP) and trains a Random Forest machine learning model using received CSI to classify human movement into “movement” and “no activity”, enabling the detection and monitoring of people.

However, most of the studies using Wi-Fi CSI that were introduced earlier do not involve wireless signals passing through multiple walls. Wang et al. [37] introduce several studies that use Wi-Fi CSI to detect people and discuss whether this can be applied to experiments that involve passing through walls. These studies demonstrate the use of CSI to detect people and track their movements when passing through 1–2 walls. However, the discussion is limited, as they do not delve into the impact of factors such as wall material, person position, and movement type on the detection performance. Similarly, research on technologies such as BLE, UWB, and LoRa also lacks consideration of scenarios with multiple walls and has limitations in terms of performance. There are a few studies among them that utilize signals passing through two or more walls for person detection [33,34]. Detecting people inside a building from the outside can be challenging due to wireless signal transmission through exterior and interior walls, as well as obstacles. When signals pass through objects, refraction occurs, influenced by the refractive index of the medium they traverse [38]. Transmitting signals between exterior walls involves thick mediums with obstacles, resulting in significant refraction and signal attenuation. While previous research using Wi-Fi or ultra-wideband signals has low distance errors for people detection due to limited signal penetration, it becomes challenging to obtain accurate results in environments with substantial attenuation, as described above.

Therefore, in this study, we used 433 MHz frequency signals, which have relatively high signal penetration, for detecting people. We attempted to measure the RSSI outside the building using low-power, long-range communication 433 MHz frequency transceivers. This RSSI value is a relative measurement value, so the range varies depending on each chip supplier; however, in this study, it ranged from approximately −60 to 105 dBm. As the distance between the transmitter and receiver increases and the interference on the signal increases, the RSSI value decreases, and the signal strength weakens.

## 3. Data Analysis Algorithms

### 3.1. RSSI Variance Analysis Algorithm

We used variance and mean analysis algorithms for data obtained through experiments to discriminate a person only by RSSI values while passing an intersection between the transmitter and the receiver [23].
(1)$$Var\left(x\right)=\frac{1}{n}\sum_{i=1}^{n}{\left(x_{i}-\mu \right)}^{2}$$
(2)$$\mu_{current}=\alpha *\mu_{old}+\left(1-\alpha \right)*{RSSI}_{current}$$

Equation (1) determines the variance in the RSSI value of the current packet, where *n* is the size of the slide window and $x_{i}$ is the RSSI value of the current packet. *μ* is an exponentially weighted moving average for RSSI values up to the present packet, and is obtained by Equation (2). The weight factor, *α*, is set to α=0.9 so that the old average RSSI value has an influence of 90% on the weighted average. The variance is obtained by collecting the most recent RSSI value by the size, *n*, of the sliding window, and the RSSI variance is obtained for packets received by a receiver for one second, considering that the time for a person to pass through the monitoring area is 0.3–0.5 s [23].

A threshold, $T_{v}$, is used as an element that determines the state of human detection in a variance analysis algorithm. If the variance at a certain point exceeds the threshold $T_{v}$, it is determined that a person has been detected and the state is indicated as 1. Conversely, if the variance is lower than $T_{v}$, it is judged that there are no people, and the state is indicated as 0. If the $T_{v}$ is too low or too high, it is possible to detect it incorrectly, so a reasonably balanced $T_{v}$ value should be found throughout the test. Through some experiments, the result was derived by setting n=10 and $T_{v}=3\;\mathrm{dBm}$, which is at the balance point for human detection.

The following experiment was conducted for five minutes to show the variance algorithm performance in a simple LOS environment. The transmission rate of the packet was one packet per 100 ms, and a total of 3000 RSSI values were analyzed. One experimental scenario is detailed in Table 1. From the RSSI values obtained in these experiments, variance is obtained using Equations (1) and (2).

A state graph, as shown in Figure 2a, appears when a state of human detection is drawn through a variance analysis with the threshold $T_{v}=3\;\mathrm{dBm}$. Using this variance value, we can obtain a state graph, as shown in Figure 2b, which indicates whether a person is present or not. It can be observed that a person is detected (state = 1) between 60 and 120 s, and at 180 and 240 s. However, during the period from 180 to 240 s, the algorithm fails to detect a person who is present but stationary (state = 0).

The biggest problem with the RSSI variance analysis algorithm is that it cannot detect a person based on variance if the person is stationary within the detection range, as the variance value does not change. In the previous experiment, while a person moving between the transmitter and receiver was detected between 60 and 120 s, the algorithm failed to detect a person standing between the transmitter and receiver during packet transmission between 180 and 240 s. To address this, the RSSI average analysis algorithm must be applied.

### 3.2. RSSI Mean Analysis Algorithm

The RSSI average analysis algorithm is suitable for detecting stationary people between the transmitter and receiver, and can detect people based on the attenuation of RSSI values. Attenuation is represented as the difference between the exponentially weighted moving average *μ* and the current RSSI value, calculated as $\;(\mu -{RSSI}_{current})$. If the absolute value of this attenuation is above a certain threshold, it predicts that a person has been detected. The exponentially weighted moving average, *μ*, at a specific point used in the attenuation calculation is calculated using Equation (2), and unlike the RSSI variance analysis algorithm, this *μ* value is not updated in events where a person is detected. Thus, the *μ* value only changes in situations where the environment changes (e.g., when obstacles are added or removed between the transmitter and receiver), and remains fixed otherwise. To distinguish the presence of a person using attenuation values, an experiment is conducted to designate pairs of coordinates (${RSSI}_{current},\;attenuation$) corresponding to the current RSSI value and its corresponding attenuation value. Linear regression analysis is then performed on these pairs of coordinates.

Figure 3a shows a linear regression graph drawn using 50 pairs of ${RSSI}_{current}$ and its corresponding attenuation values obtained from an experiment where a person is passing by. During the experiment, the person passes through the detection range, stays still within the detection range, or is outside the detection range. Random samples of 50 pairs of data obtained from the experiment were selected as pairs of coordinates. If the detection environment does not change, the *μ* value is also fixed, and since the *μ* of the pair of coordinates (${RSSI}_{current},attenuation$) does not change, all coordinates are plotted on the linear regression graph as pairs of coordinates. The linear regression equation for the first-degree function when the environment does not change is expressed by Equation (3). Here, m is the slope of the linear regression graph and c is the y-intercept of the graph.
(3)$$attenuation=m*{RSSI}_{current}+c$$

In the absence of a person, the attenuation’s absolute value is generally 0. However, external noise unrelated to a person can introduce slight deviations from zero. To account for this noise, a filter is used to detect a person when the absolute value of attenuation reaches or surpasses a predetermined constant value,$\;T_{g}$. This value is typically determined through experimentation, finding the threshold that effectively distinguishes between the presence and absence of a person while considering the impact of noise. This value is usually smaller than the attenuation when a person moves. $T_{g}$ is represented by a horizontal line parallel to the x-axis in Figure 3b.

When there is a change in the detection environment, the value of the exponential moving average, *μ*, becomes variable. Consequently, the pair of values (${RSSI}_{current},attenuation$) deviates from the linear regression graph described by Equation (3), affecting the graph itself. In a stable environment, *μ* remains fixed and the slope, *m*, is equal to −1. When the environment changes and a new linear regression graph is generated, the slope, *m*, remains as −1 while only the y-intercept value changes. The linear regression graph shifts by a certain amount along the y-axis, and the maximum magnitude of this shift in the y-intercept is denoted as a constant value, β. By specifying values for $T_{g}$ and β, it becomes possible to define the region where the presence or absence of a person can be distinguished. This region is referred to as the $T_{m}$ area, enclosed by three linear function graphs, $T_{g}$, $T_{d1}$, and $T_{d2}$, as illustrated in Figure 3b. $T_{d}$ is defined by Equation (4), which is a filter considering environmental changes, and is expressed as $T_{d1}$ when + β is added to Equation (3), and as $T_{d2}$ when − β is added.
(4)$$T_{d}=m*{RSSI}_{current}+c\pm \beta$$

In the mean analysis algorithm, similar to the threshold value, $T_{v}$, utilized in the variance analysis algorithm, the threshold parameters, $T_{g}$ and β, are determined during the parameter-setting stage by combining them with suitable values. The objective is to find the combination that yields the highest accuracy of the algorithm by comparing the predicted data with the actual data. The state of person detection is binary, indicating one of two states. If the attenuation falls within the $T_{m}$ area, it is determined that a person has been detected, and the state is denoted as 1. Conversely, if the attenuation lies outside the $T_{m}$ area, it is determined that there is no person, and the state is denoted as 0. In the verification stage, the F1-score is calculated to evaluate the algorithm’s performance.

Precision is a metric that represents the number of true positive results (i.e., the number of times the analysis algorithm correctly identifies an observed event as positive) divided by the total number of positive results returned by the algorithm, which includes true positives and false positives (i.e., the number of times the analysis algorithm incorrectly identifies an observed event as positive).

Recall, on the other hand, is a metric that represents the number of true positive results divided by the total number of actual positive instances in the dataset, which includes true positives and false negatives (i.e., the number of times the analysis algorithm fails to identify an observed event as positive).

In summary, precision measures the accuracy of positive predictions made by the analysis algorithm, while recall measures the algorithm’s ability to identify all positive instances in the dataset. Both metrics are important when evaluating the performance of a binary classification algorithm.

The F1-score is a single metric that combines both precision and recall, and it can be calculated using the following equation:$$F1\;score=2*\;\frac{precision*recall}{precision+recall}$$

The F1-score is a harmonic mean of precision and recall, and it ranges from 0 to 1, with a higher score indicating better performance. It is a useful metric for evaluating the overall effectiveness of a binary classification algorithm because it balances both precision and recall, taking into account both false positives and false negatives. A higher F1-score indicates a better balance between precision and recall, while a lower score indicates that the algorithm is either biased towards precision or recall.

Section 5 presents the results of experiments conducted at a specified location where there is at least one wall separating the transmitter and receiver. The laser transmitters and receivers are employed to collect actual human movement data in order to calculate accuracy. The laser receiver detects the signals emitted by the laser transmitter as a person moves between the RF transmitter and receiver. By utilizing the laser-based system, the presence or absence of a person can be accurately determined, and these data are then compared with the RSSI values obtained through RF signals. The combination of laser-based detection and RF signal measurements provides a comprehensive dataset for analyzing and evaluating the accuracy of the system’s detection capabilities. Precision and recall will be calculated to evaluate the experiment’s performance, and the F1-score will be computed to determine its accuracy.

## 4. Experimental Overview

### 4.1. The Direction of the Experiment

In some recent studies [23,24,25,27,29,34], methods of determining the presence or absence of a person through RSSI values were used. However, most of these studies used high-frequency signals; this method is difficult to use in special situations due to low wall permeability. This paper proposes a new method for indoor human detection using low-power 433 MHz frequency wireless signals, which can penetrate walls more effectively than high-frequency signals. This study aimed to verify whether RSSI values can be used to detect human presence even in environments where the frequency band is lower than Wi-Fi, and whether the proposed algorithm is effective in detecting people behind multiple walls from outside the building. We conducted experiments using the STM32 board (Geneva, Switzerland) and 433 MHz frequency band to check if the RSSI values are affected by human movement. They aimed to verify whether detection using RSSI values is possible even in an environment where the frequency band is relatively lower than Wi-Fi. We verified the practicality of detecting people from outside the building even when there are two or more walls between the transmitter and receiver. For our experiments, we manufactured a PCB using the STM32F103C8T6 and CC1120 wireless communication modules (Dallas, TX, USA), which use the ARM Cortex-M3 MCU shown in Figure 4 as a transceiver for human detection.

In the experiments, the CC1120 wireless communication module was used with specific settings, as outlined in Table 2. The module operated at a frequency of 433 MHz, and it is possible to configure the following parameters:Bandwidth: the maximum bandwidth can be set to 200 kHz.Bit Rate: the maximum bit rate can be set to 100 kbps.Tx Power: the maximum power can be set to 15 dBm.
sensors-23-06280-t002_Table 2Table 2CC1120 module setup.CC1120 Module SetupCarrier Frequency433 MHzTx Power10 dBmBit Rate1.2 kbpsModulation2-FSKBandwidthAntenna8.5 KHz12dbi (DexMRtic, Model: SD-10)

It is worth highlighting that adherence to domestic regulations in South Korea is crucial when utilizing the 433 MHz frequency. In this study, we initially employed RF settings that aligned with the regulated parameters specified by Korean policy. However, we are now interested in exploring additional settings beyond these limitations. These restrictions are as follows:Bandwidth: the bandwidth should be set to 8.5 kHz or less.Bit Rate: within this bandwidth, the bit rate can be set to 1.2 kbps to prevent data corruption.Tx Power: the transmission power should be set to 10 dBm or less.

Regarding the modulation format, the CC1120 module offers various options, including 2-FSK, 2-GFSK, ASK/OOK, 4-FSK, and 4-GFSK. For the narrowband settings in the experiments, 2-FSK modulation was selected. This choice is recommended by the manufacturer for optimal sensitivity in such scenarios. Although the CC1120 module does not provide detailed channel information like Wi-Fi, it does offer the RSSI for each received packet. The next consideration is the impact of multipath propagation. Multipath propagation occurs when wireless signals are reflected, refracted, and diffracted by obstacles in the environment, resulting in multiple copies of the signal arriving at the receiver at different times and amplitudes. This can cause signal fading, interference, and errors in signal detection and decoding. To enhance the accuracy of human detection by mitigating the effects of multipath propagation, multiple separated receiver modules are employed for comparative analysis. As detailed in Section 5, three distinct receiver modules were utilized, with each module receiving packets containing sequence numbers from a single transmitter. The sequence numbers from each receiver serve as a temporal axis for predicting human detection and generating datasets. However, the current hardware limitations, such as the number of available serial communication ports on the board, necessitated the use of three separate receiver modules. For future considerations, we plan to explore the possibility of implementing a single board with three receiving RF modules. This approach offers the advantages of a consolidated system while maintaining the benefits of multiple receiving points for enhanced accuracy in human detection.

### 4.2. Data Analysis

We conducted several experiments to determine the effects of factors such as antennas, signal strength, the presence or absence of obstacles, and distance on the RSSI values. All experiments were conducted in corridors and rooms in an 11-storey building, and omnidirectional antennas were used. The transmitter transmitted packets to the receiver at 0.2 s intervals, and a person walked back and forth between the transceivers for 180 s. Four experiments were conducted with different settings, including varying the TX power, the position of the transmitter and receiver, and the distance between them. There was a wall between the transmitter and receiver when the receiver was in the laboratory. The experimental settings are shown in Table 3.

The experimental results in Figure 5 show that the RSSI value remained stable when there was no human present, but when a person walked between the transceivers, the RSSI value was affected. The signal strength was attenuated by the wall, but the affected signal showed a noticeable pattern that could be distinguished from other patterns regardless of the wall’s presence.

One notable distinction of this system compared with other protocols lies in the range of signal attenuation. In the case of the 433 MHz frequency band, as depicted in Figure 5, the attenuation of RSSI when a person is present is relatively small, around ±1~2 dBm. This attenuation is significantly lower compared with the approximately ±10 dBm attenuation observed in Wi-Fi measurements at frequencies such as 2.4 GHz and 5 GHz [14]. The lower frequency of the 433 MHz signal enables it to traverse obstacles more easily than higher-frequency Wi-Fi signals, resulting in reduced signal attenuation. Consequently, our system can accurately detect the presence of a person even when facing ±1~2 dBm attenuation. However, it should be noted that this attenuation is dependent on the experimental environment, and our approach requires a setup phase to determine the threshold accordingly. Furthermore, by employing multiple separated receivers, we can mitigate the impact of RSSI attenuation range variations among different boards. This approach ensures that even if some boards exhibit smaller attenuation ranges, the system’s performance remains robust.

Furthermore, another point to consider is the scenario where multiple people pass through the detection range, rather than just one person. When comparing the scenario of one person walking through the detection range with the scenario of three people walking through, it was observed that when multiple people pass through the detection range, the RSSI attenuation is measured to be much greater. This implies that if the algorithm in Section 3 is capable of detecting one person, it should also be able to detect multiple people passing through with even greater attenuation. The difference in attenuation can be used to distinguish between scenarios where one person or multiple people pass through. However, detecting people in a smaller attenuation range is more challenging, which is why Section 5 focuses on experimenting with scenarios where only one person passes through.

## 5. Experiment and Results

This section aims to verify the accuracy of the analysis algorithm introduced in Section 3.1 and Section 3.2 through actual experiments, specifically in scenarios where individuals move around randomly. The experiments are conducted in three different scenarios. First, a single wall is placed between the transmitter and receiver, and a person walks along a predetermined scenario. Second, two walls are placed, and a person walks randomly. Finally, the transmitter and receiver are placed outside the building, and a person walks randomly inside the building. The transceiver setup for each experiment is given in Table 4.

Before conducting the experiments, it was essential to address the issue of detection range. At present, the measurement method entails an individual traversing a designated line positioned between the laser transmitter and receiver in order to obtain the measurements. However, the received RSSI value will be affected *n* seconds before and after the person enters the laser line. This period should be considered as a human detection event. To address this issue, the detection range is extended to include the 2–4 s before and after the human is detected in the ground truth data from the laser receiver. The extended detection range depends on the distance between the transmitter and receiver. To account for the variation in the detection range depending on the location of the experimental device, the offset is obtained and stored in the initial parameter setup phase, which narrows the gap between the predicted data obtained through the analysis algorithm and the actual data. This offset is applied to the predicted data in the verification phase. To ensure optimal performance in the specific environment of the experiment, it is recommended to choose the receiver board that provides the best performance. Thus, for each experiment, the receiver board may be replaced accordingly.

As reported in Section 5, all experiments followed a similar process. In each experiment, the transmitter sent packets every 0.2 s, and each of the three receivers collected a total of 5000 packets over a duration of 1000 s. During the experiment, a single participant randomly moved back and forth. The experiment consisted of an initial setup phase to set the values of linear regression and $T_{v}$, $T_{g}$, and β, followed by a validation phase to calculate the accuracy. In the setup phase, the algorithm from Section 3 was applied to 70% (3500 packets) of the measurement data to obtain parameter values (e.g., $T_{v}$, $T_{g}$, β, offset, etc.). In the validation phase, these parameter values were applied to 30% (1500 packets) of the measurement data to compare the predicted data with the actual measured data and calculate the accuracy. For experiments B and C (excluding Experiment A), this process was repeated twice. Experiment B had two phases, denoted as B1 and B2, while Experiment C also had two phases, denoted as C1 and C2.

In the conducted experiments, the detection algorithm was executed offline, but the RSSI value of each received packet on each module with a packet ID was transmitted to the server in real time. This transmission was facilitated using Wi-Fi on each receiver module to send the RSSI information to a mobile Access Point (AP). The AP acted as an intermediary, relaying the data to the server via an LTE connection for subsequent processing and analysis. All experimental data were saved on the server for further reference and analysis. In our initial implementation, we utilized three separate modules. However, to optimize the deployment of the detection system, an alternative approach involves integrating three receiver modules into a single board with at least three serial communication I/O ports to accommodate the modules. By consolidating the modules onto a single board and leveraging its capabilities, the detection system can be efficiently executed on the board itself in real time, eliminating the need for separate modules. This integration enhances the system’s performance and streamlines its operation.

### 5.1. Experiment A: Experiment with a Wall

In Experiment A, the setup phase involved conducting the experiment in the same location as depicted in Figure 6. A 5 cm steel door was positioned between the transmitter and receiver. The purpose of this experiment was to evaluate the system’s performance in detecting humans located inside the building while dealing with the obstruction posed by the steel door.

The linear regression function for Experiment A is shown in Figure 7 and Table 5. The graph in Figure 7 provides a rough estimate of the attenuation range when a person is present. To illustrate this, let us consider Figure 8, which represents the time–RSSI graph for Board 3 in Figure 7. The solid graph represents the values of RSSI, while the scatter plot overlaid on it indicates the actual presence of a person at specific times. When there is no person, the RSSI remains relatively stable with an average of −67 dBm. However, when a person is present (as indicated by the scatter plot), the RSSI fluctuates between −72 dBm and −65 dBm. Analyzing Board 3 in Figure 7 specifically, when there is no person, the RSSI values are −68 dBm and −67 dBm, labeled as “human absence”. When there is a person, the RSSI values range from −72 dBm to −65 dBm, labeled as “human presence”. Therefore, when there is only the “human presence” label (|attenuation| ≥ 2 dBm) present, we can infer that there is always a person present.

Using the slope and y-intercept obtained from the linear regression function, we substituted them into the mean analysis algorithm during the setup phase to obtain the maximum F1-score of the variance–mean analysis algorithm, and the corresponding values of $T_{g}$, β, and offset. Table 6 summarizes the maximum F1-scores of the variance–mean analysis algorithm, and their corresponding parameters for each board, obtained from the dataset during the setup phase of Experiment A.

After the setup phase, the obtained parameters were used to generate predicted data, which were then compared with the actual measured data to calculate accuracy using the F1-score in the validation phase. During the verification phase, the parameters obtained in the setup phase were used to generate a state graph that predicted the presence or absence of a person over time. State values of 1 and 0 indicated the presence and absence of a person, respectively. Additionally, a scatter plot representing the ground truth data was superimposed on the predicted solid line graph. The ground truth data represent the actual measurement data when a person was present. Figure 9 shows an example state graph obtained in the verification phase.

The result of the verification phase is a state graph, as shown in Figure 9. Figure 9 provides guidance on how to interpret the results of predictions and compare them with the actual data. It helps in understanding the relationship between the system’s predictions and the ground truth. By default, the predicted solid line graph, which predicts the presence or absence of a person over time, had state values of 1 and 0. When the state was 1, it meant that the system predicted that there was a person at that time, and when the state was 0, it meant that the system predicted that there was no person at that time. The additional scatter plot on the predicted solid line graph represents the actual measurement data where a person was present, labeled as the ground truth.

The state graph generated during the verification phase can be categorized into four main cases. As shown in Figure 9, these cases can be labeled as follows:Case A represents the scenario where the system accurately predicts the presence of a person at a specific time.Case B indicates the situation where the system correctly predicts the absence of a person at a certain time.Case C, where the system predicts the presence of a person at a certain time, but in reality, there is no person present.Case D case occurs when the system predicts the absence of a person at a certain time, but in reality, there is a person present.

Cases A and B correspond to correct predictions, while cases C and D denote incorrect predictions. The F1-score in Table 7 is the result of Experiment A, and the time–state graph of the verification phase is shown in Figure 10. All boards achieved F1-scores above 0.9 with no outliers in Experiment A.

### 5.2. Experiment B: Experiment with Two Walls

Experiment B was performed under identical conditions to Experiment A, except for the relocation of the receiver. The location for Experiment B involved placing the transmitter and receiver in separate rooms with a corridor in between, as depicted in Figure 11. The signal had to pass through two 5 cm steel doors to reach the receiver; the purpose of this experiment was to evaluate the system’s ability to detect people in the presence of obstacles. In the additional Experiment B2, a person moved randomly back and forth between the transmitter and receiver, occasionally stopping in between, to test whether the system could detect stationary individuals within its detection range. The results of the linear regression analysis in Experiment B are summarized in Table 8. The F1-score in Table 9 is the result of applying the parameters obtained in the setup phase to the verification phase data of Experiment B1. The time–state graph of the verification phase is shown in Figure 12.

The peculiarity is that the mean analysis algorithm on Boards 4 and 6 falsely detected or failed to detect some events due to noise. As there were differences and errors in the performance of each board, including hardware malfunctions, three boards were selected as receivers and analyzed using both the mean and variance analysis algorithms. The results from the three analyzed boards were then combined to determine the final state by selecting the state that had two or more boards with the same state value at each time point. Experiment B2 was conducted to confirm whether people who were standing within the detection range could be detected. This experiment was carried out under the same conditions as B1; the experimenter moved randomly, and occasionally paused. The F1-scores in Table 10 are the result of applying the parameters obtained in the setup phase to the verification phase data of Experiment B1. The time–state graph of the verification phase is shown in Figure 13.

The anomaly observed in Experiment B2 is that the mean analysis algorithm was able to detect people standing still in the 140–160 s and the 260–290 s intervals, while the variance analysis algorithm failed to detect them. This is because the variance analysis algorithm relies on changes in the variance value to detect the presence of a person, which occurs when a person is moving. However, when a person is stationary, the variance value remains constant, and therefore, the algorithm fails to detect them. On the other hand, the mean analysis algorithm uses mean values to detect the presence of a person. Even when a person is stationary, the mean value is lower than when there is no person present, allowing the algorithm to detect stationary individuals. However, as an exception, Board 6 failed to detect stationary individuals even with the mean analysis algorithm. Figure 14 presents the dataset of Board 6 in Experiment B2, where the average RSSI when no person is present and the RSSI values during the intervals when a person is stationary are identical, resulting in an RSSI attenuation of 0 and causing the algorithm to fail in detection.

### 5.3. Experiment C: External Walls and Internal Obstacles within the Building

The receiver’s placement was altered in Experiment C while maintaining the same conditions as Experiment B. The final experiment, Experiment C, involved placing the transmitter–receiver outside the building’s two exterior walls, as depicted in Figure 15. Between the transmitter and receiver, there were three gypsum walls with a thickness of 20 cm and one concrete pillar with a thickness of 100 cm as interior walls. Additionally, there were two concrete pillars with a thickness of 100 cm and windows with a thickness of 5 cm on either side of the pillars as exterior walls. Figure 16 presents a photo depicting the equipment setup for better understanding. The straight path from the receiver to the transmitter follows the order A, B, C, as shown in Figure 16; the distance between the transmitter and receiver is 20 m. Figure 16A represents the receiver outside the concrete pillar, and Figure 16B represents the corridor inside the window of Figure 16A, where the laser transmitter–receiver is installed. After passing through a large room with four inner walls, Figure 16C represents the transmitter installed outside the concrete pillar on the opposite outer wall.

The experimenter moved randomly up and down the path between the transmitters and receivers, performing Experiments C1 and C2, and then added a stationary situation similar to Experiment B2 for Experiment C2. The results of the linear regression analysis in Experiment B are summarized in Table 11.

The F1-scores in Table 12 are the results of Experiment C1; the time–state graph of the verification phase is shown in Figure 17.

The last experiment, Experiment C2, was the same as Experiment B2, where the experimenter moves randomly and sometimes stops, and the remaining conditions are the same as C1. The F1-scores in Table 13 are the results of Experiment C1; the time–state graph of the verification phase is shown in Figure 18.

As depicted in Figure 19, the RSSI data obtained from indoor Experiments A and B, as well as the data from outdoor Experiment C, exhibit slightly different characteristics. The experimental results indicate that even a small change in RSSI can be effective in detecting a human presence in Experiment B (Figure 19a). However, in Experiment C (Figure 19b), where multiple walls were present, the RSSI values often fluctuated even in the absence of people. Nevertheless, when there was attenuation amidst these fluctuations caused by a person passing by, the system can successfully detect the presence of a person.

The mean analysis algorithm usage results in Experiments A to C are mostly 0.9 or higher. This suggests that the mean analysis algorithm is effective in detecting the presence of people in an area. Table 14 summarizes the F1-score of the variance and mean analysis algorithms in each validation phase of Experiments A to C, combining the three boards. This information may be useful for evaluating the performance of these algorithms and for comparing their effectiveness in different experimental settings.

When averaging across the items in Experiments A through C, the mean F1-score for the variance analysis algorithm is 0.880 and the mean F1-score for the mean analysis algorithm is 0.977. This suggests that, on average, the mean analysis algorithm performs better than the variance analysis algorithm and the mean analysis algorithm has a very high level of accuracy, with its F1-score being almost comparable to 100%. These findings may be useful for evaluating the effectiveness of these algorithms in detecting the presence of people in different experimental settings.

During Experiment C, it was observed that despite the existence of multiple reliable human detection systems [8,16], the RF frequencies employed by these systems demonstrated limited capability in penetrating the building. The high loss experienced by the RF signals rendered it unsuitable for implementing similar techniques in the depicted environment, as shown in Figure 15.

## 6. Conclusions

This paper presents a novel human detection system that utilizes a 433 MHz frequency signal outside a building to detect human presence inside. Using a low-power, long-distance communication signal at 433 MHz, the system overcomes attenuation issues faced by existing RF devices, enabling detection from outside the walls. The paper proposes two data analysis methods, variance and mean analysis algorithms, for processing the collected data. The mean analysis algorithm achieves higher accuracy and can detect stationary individuals. In the validation phase, the variance analysis algorithm achieved an average F1-score of 0.880, while the mean analysis algorithm achieved an average F1-score of 0.977. A drawback of the system is the requirement for a new setup phase for all boards when performing detecting in a new location, resulting in repetitive setup. The objective was to simplify or eliminate the setup and configuration process across multiple locations, enhancing user-friendliness and system efficiency.

If a proposed system can be used immediately after installation, it is expected to be useful for detecting people in disaster situations, such as fire scenarios, where it is difficult to install the device in advance and obtain building parameters. Overall, this paper proposes a novel approach to human detection inside the building using a low-power, long-distance communication signal; the results demonstrate the potential effectiveness of the proposed system.

## Figures and Tables

**Figure 1 sensors-23-06280-f001:**
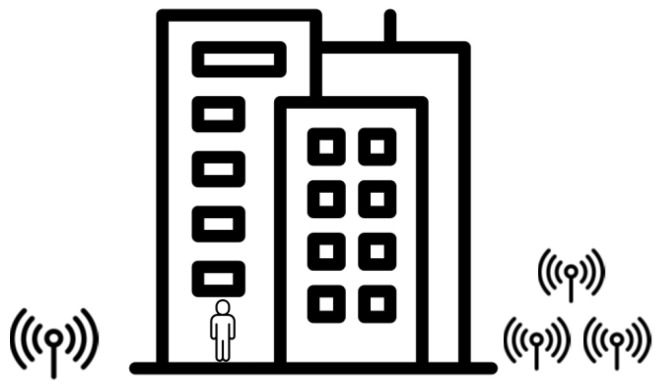
Situation of indoor people detection from outside the building.

**Figure 2 sensors-23-06280-f002:**
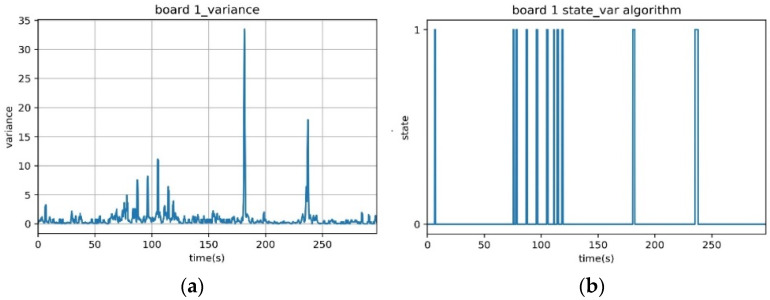
Results of the performance analysis experiment for the RSSI variance analysis algorithm. (**a**) Variance graph, (**b**) Expected presence of a person graph.

**Figure 3 sensors-23-06280-f003:**
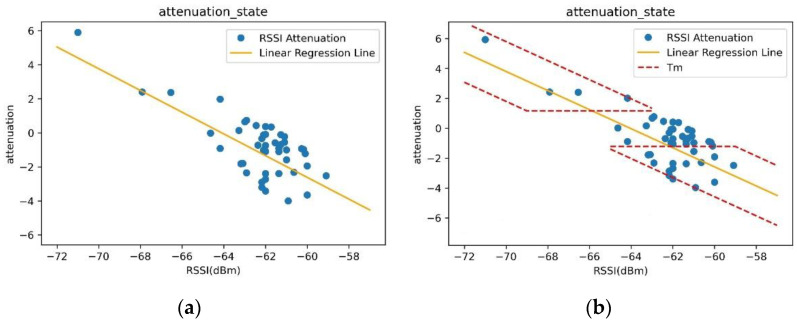
Left panel (**a**) scatter plot and linear regression graph of the current RSSI value (x-axis) and attenuation (y-axis) and right panel (**b**) *T_m_* area added to determine the presence or absence of a person.

**Figure 4 sensors-23-06280-f004:**
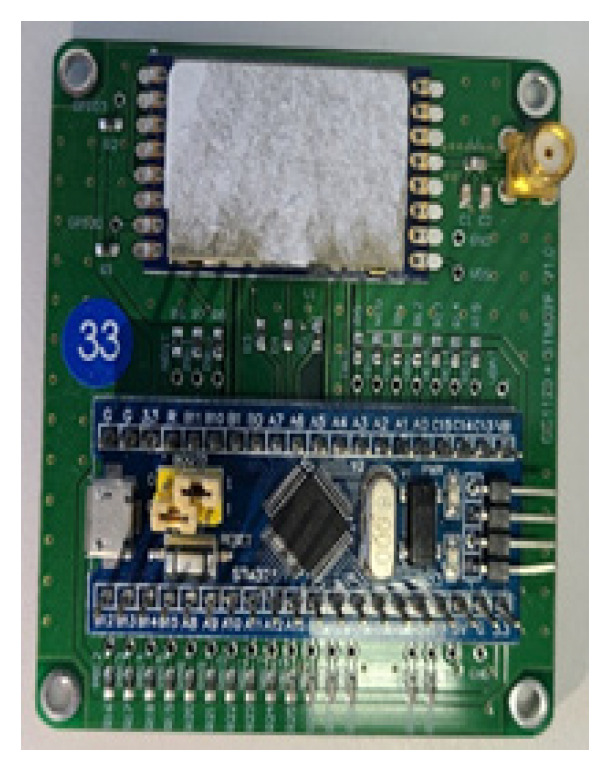
Transceiver configured with STM32F103 + CC1120.

**Figure 5 sensors-23-06280-f005:**
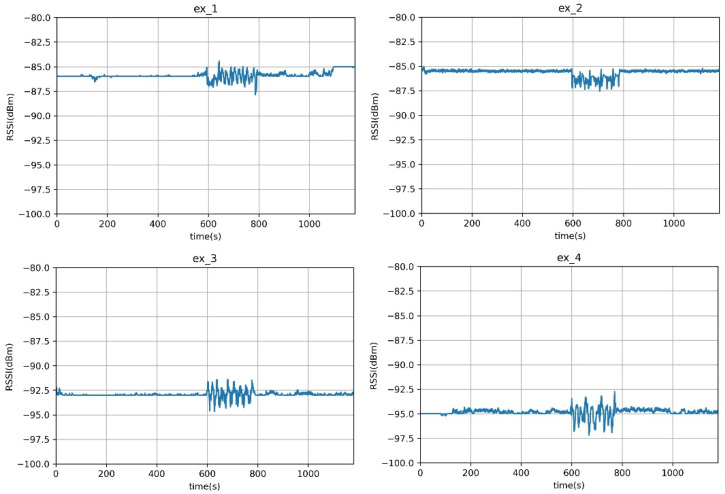
Experimental results: x-axis: time (s); y-axis: RSSI value. Ex_1 and Ex_2: LOS; Ex_3 and Ex_4: NLOS.

**Figure 6 sensors-23-06280-f006:**
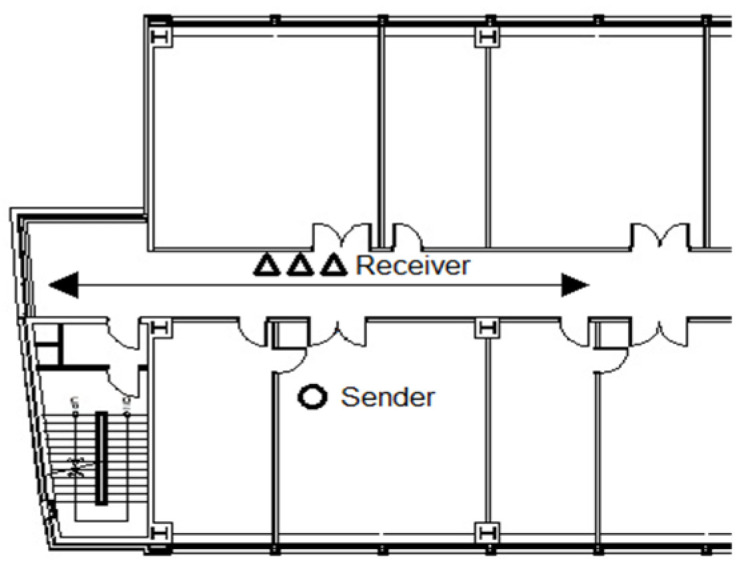
Diagram of the transmitter–receiver placement and path of travel for Experiment A.

**Figure 7 sensors-23-06280-f007:**
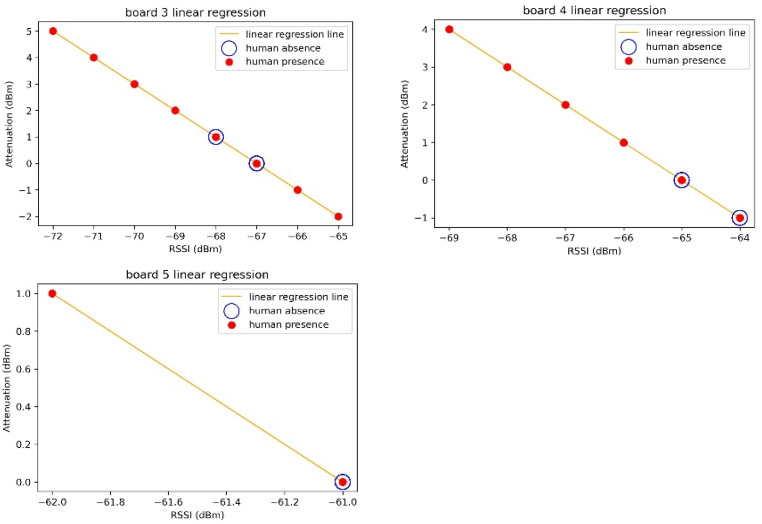
Linear regression graphs for Boards 3, 4, and 5.

**Figure 8 sensors-23-06280-f008:**
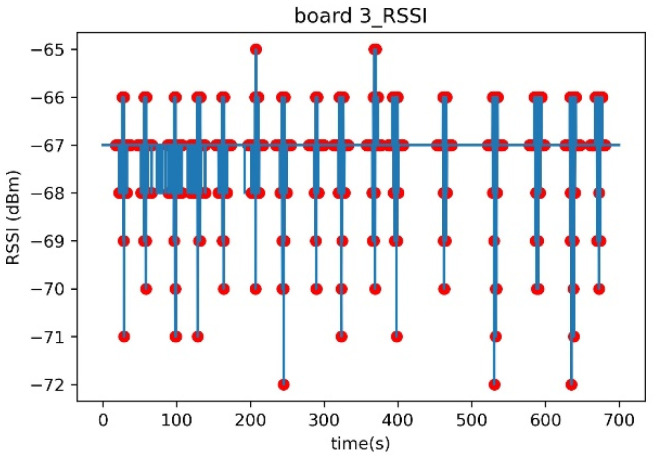
Time–RSSI graph for Board 3.

**Figure 9 sensors-23-06280-f009:**
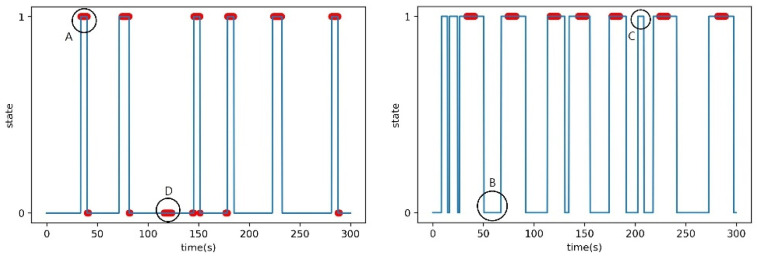
Examples of the state graph in the verification phase. A: True Positive, B: False Negative, C: False Positive, D: True Negative.

**Figure 10 sensors-23-06280-f010:**
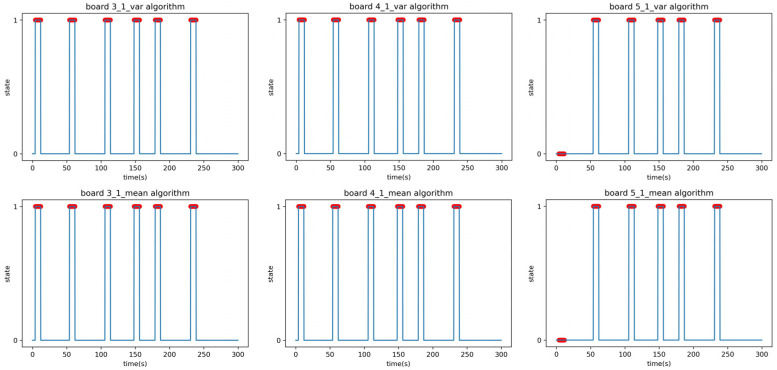
Time–state graphs of the verification phase of Experiment A: (**top**) variance analysis algorithm; (**bottom**) mean analysis algorithm.

**Figure 11 sensors-23-06280-f011:**
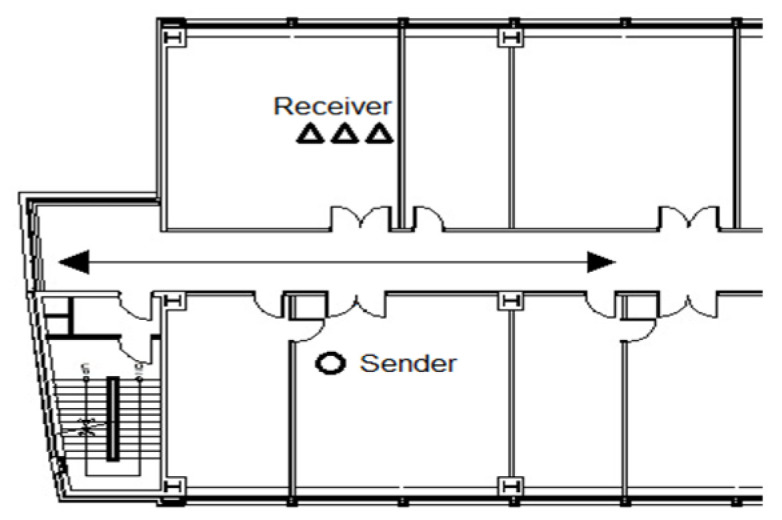
Diagram of the transmitter–receiver placement, and the path of travel for Experiment B.

**Figure 12 sensors-23-06280-f012:**
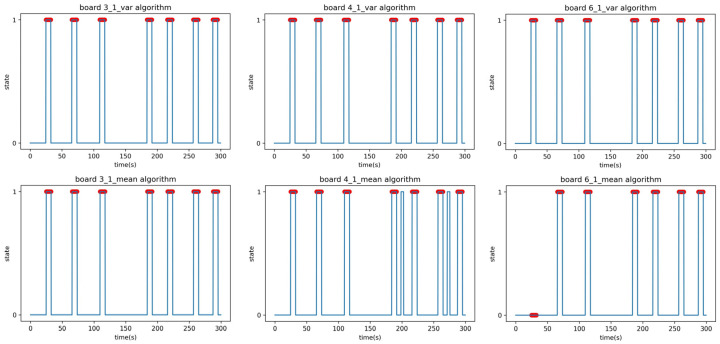
Time–state graphs of the verification phase of Experiment B1: (**top**) variance analysis algorithm; (**bottom**) mean analysis algorithm.

**Figure 13 sensors-23-06280-f013:**
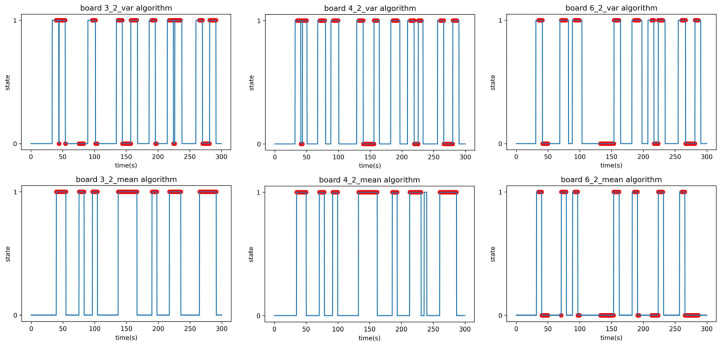
Time–state graphs of the verification phase of Experiment B2 (adding stationary positions): (**top**) variance analysis algorithm; (**bottom**) mean analysis algorithm.

**Figure 14 sensors-23-06280-f014:**
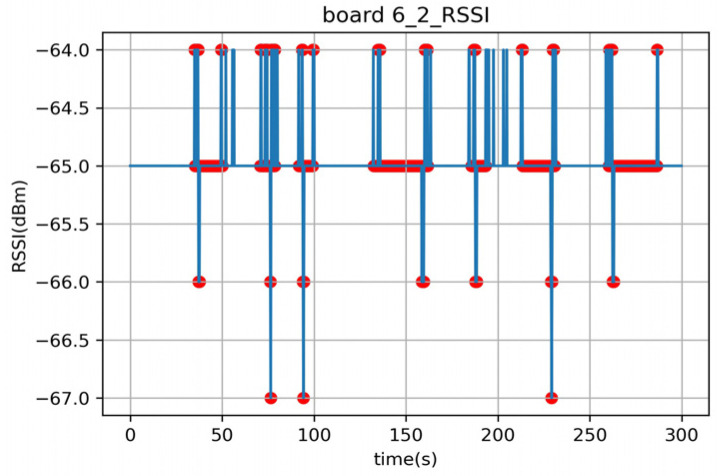
Time–RSSI graph of the verification phase of Board 6 in Experiment B2.

**Figure 15 sensors-23-06280-f015:**
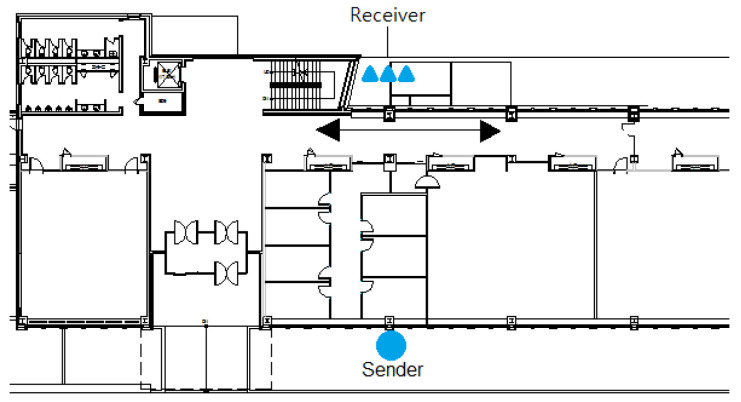
Diagram of the transmitter–receiver placement and path of travel for Experiment C.

**Figure 16 sensors-23-06280-f016:**
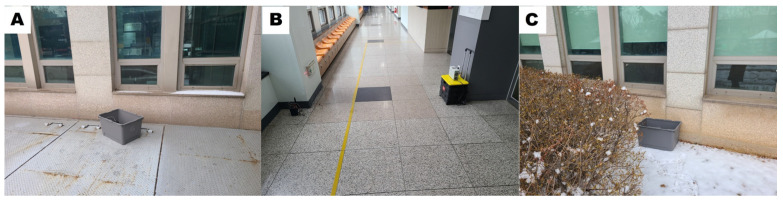
Layout of the equipment used in Experiment C.

**Figure 17 sensors-23-06280-f017:**
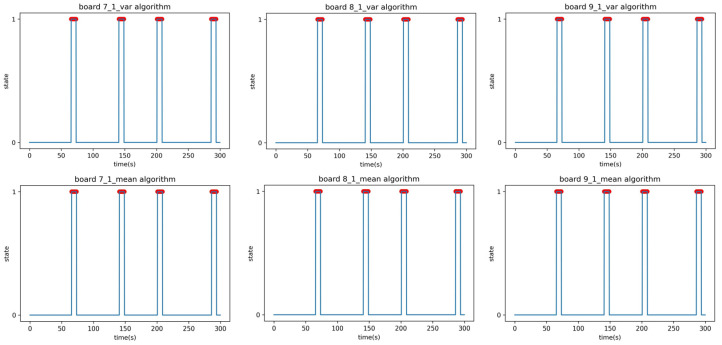
Time–state graphs of the verification phase of Experiment C1: (**top**) variance analysis algorithm; (**bottom**) mean analysis algorithm.

**Figure 18 sensors-23-06280-f018:**
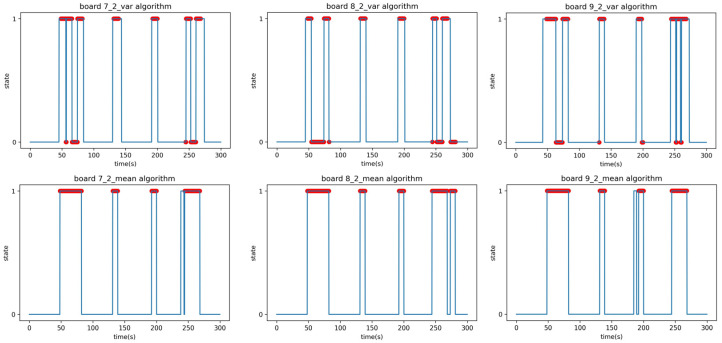
Time–state graphs of the verification phase of Experiment C2: (**top**) variance analysis algorithm; (**bottom**) mean analysis algorithm.

**Figure 19 sensors-23-06280-f019:**
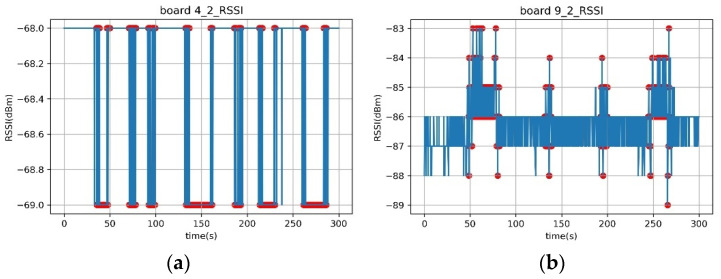
Time–RSSI graphs of the verification phase of Experiments B and C. (**a**) Experiment B2, Board 4, (**b**) Experiment C2, Board 9.

**Table 1 sensors-23-06280-t001:** Experimental scenario to apply variance and mean analysis algorithms.

Time	0~60 s	60~120 s	120~180 s	180~240 s	240~300 s
Scenario	No human	A person periodically passing between the transceiver	No human	A person standing between the transceiver	No human

**Table 3 sensors-23-06280-t003:** Content of various element analysis experiments affecting signal.

	TX Power	Transmitter Position	Receiver Position	Distance
Ex_1	15 dBm	Corridor	Corridor	30 m
Ex_2	6 dBm	Corridor	Corridor	20 m
Ex_3	6 dBm	Corridor	Laboratory	10 m
Ex_4	15 dBm	Corridor	Laboratory	30 m

**Table 4 sensors-23-06280-t004:** Transceiver setup locations for each experiment.

	Transceiver Setup
Ex_A	Experiment with a single wall (Section 5.1)
Ex_B	Experiment with two walls (Section 5.2)
Ex_C	Experiment with multiple inside walls: the transmitter and receiver are placed outside the building with multiple interior walls between them, and a person walks randomly inside the building (Section 5.3)

**Table 5 sensors-23-06280-t005:** Linear regression analysis results—Boards 3, 4, and 5.

	Slope (*m*)	y-Intercept (c)
Board 3	−1.000	−67.000
Board 4	−1.000	−65.000
Board 5	−1.000	−61.000

**Table 6 sensors-23-06280-t006:** Experiment A setup phase results.

	F1-Score (Variance)	*T_v_*	F1-Score (Mean)	*β*	*T_g_*	Offset
Board 3	1.000	0.77	1.000	0	2	1
Board 4	1.000	0.59	1.000	0	2	1
Board 5	0.908	0.06	0.908	0	1	1

**Table 7 sensors-23-06280-t007:** Experiment A verification phase results.

	F1-Score (Variance)	F1-Score (Mean)
Board 3	1.000	1.000
Board 4	1.000	1.000
Board 5	0.908	0.907

**Table 8 sensors-23-06280-t008:** Linear regression analysis results—Boards 3, 4, and 6.

	Slope (*m*)	y-Intercept (c)
Board 3	−1.000	−70.000
Board 4	−1.000	−68.000
Board 6	−1.000	−65.000

**Table 9 sensors-23-06280-t009:** Experiment B1 verification phase results.

	F1-Score (Variance)	F1-Score (Mean)
Board 3	1.000	1.000
Board 4	1.000	0.930
Board 6	1.000	0.924

**Table 10 sensors-23-06280-t010:** Experiment B2 verification phase results.

	F1-Score (Variance)	F1-Score (Mean)
Board 3	0.670	1.000
Board 4	0.648	0.982
Board 6	0.556	0.770

**Table 11 sensors-23-06280-t011:** Linear regression analysis results—Boards 7, 8, and 9.

	Slope (*m*)	y-Intercept (c)
Board 7	−1.000	−83.000
Board 8	−1.000	−87.000
Board 9	−1.000	−87.000

**Table 12 sensors-23-06280-t012:** Experiment C1 verification phase results.

	F1-Score (Variance)	F1-Score (Mean)
Board 7	1.000	1.000
Board 8	1.000	1.000
Board 9	1.000	1.000

**Table 13 sensors-23-06280-t013:** Experiment C2 verification phase results.

	F1-Score (Variance)	F1-Score (Mean)
Board 7	0.753	0.967
Board 8	0.643	1.000
Board 9	0.795	0.971

**Table 14 sensors-23-06280-t014:** Experiment A~C verification phase maximum F1-scores.

	F1-Score (Variance)	F1-Score (Mean)
Ex_A	1.000	1.000
Ex_B1	1.000	0.930
Ex_B2	0.648	0.982
Ex_C1	1.000	1.000
Ex_C2	0.753	0.971
Average	0.880	0.977

## Data Availability

Not applicable.

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
