# Peer review of "Indoor Human Detection from a Building’s Exterior Using 433 MHz Wireless Transceivers"

_sensors, 2023, doi:10.3390/s23146280_

Round 1
Reviewer 1 Report
The authors present a study aiming at detecting human presence inside buildings using the received signal strength indicator (RSSI) information of a commercial radio frequency (RF) device.
This is a well-studied research topic, and the paper fails to provide an overview of the state-of-the-art (as relevant publications are missing). As a result, a performance comparison with existing methods is missing.
The "Introduction/Background" section needs to be reworked. All statements should be properly referenced, e.g., at line 196: "in a recent study" (this study should be referenced).
As several papers using CSI-based human detection are mentioned, with the statement it performs better (at line 89), a more detailed explanation should be given.
At line 169, it is mentioned that no studies exist to detect human presence via RF when there are walls in between the two antennas. However, this is far from correct. I believe many relevant papers are omitted from the literature/background section, including:
- "RF Based Real Time Human Motion Sensing" https://ieeexplore.ieee.org/document/9703954
- "RF System Design for Passive Detection of Humans" https://ieeexplore.ieee.org/document/8877273
- "A New RF Sensing Framework for Human Detection Through the Wall" https://ieeexplore.ieee.org/document/9940546
- "Human Sensing in Reverberant Environments: RF-Based Occupancy and Fall Detection in Ships" https://ieeexplore.ieee.org/document/9388901
- "Through-Wall Detection of Human Being's Movement by UWB Radar" https://ieeexplore.ieee.org/document/6189030
- "Sense-through-wall human detection based on UWB radar sensors" https://www.sciencedirect.com/science/article/abs/pii/S0165168415003242
- "TW-See: Human Activity Recognition Through the Wall With Commodity Wi-Fi Devices" https://ieeexplore.ieee.org/document/8515041
The statement at line 192 is wrong. Even though RSSI is not a real power measurement, it gives a power indication, and with increasing antenna separation, the path loss will be higher and the received power will be lower. The RSSI will therefore decrease.
Figure 1b is confusing. Later in the paper it is stated that this represents only a single scenario and other scenarios are considered where both transceivers are inside the building. I believe this figure represents an overview of the envisioned (final) deployment. In this case, the figure should move to the Introduction section.
Regarding the statement at line 222 that having more data points (i.e., receivers) increases the accuracy of human detection: are the receivers synchronized? Is the data from the different receivers processed jointly? It is not clear from the manuscript how the data from the different sensors is jointly processed as only a single board is mentioned.
In the presentation of measurement setup (Section 3.1) and the CC1120 module (Table 4 in Section 4), there should be a discussion on the dynamic range of the setup and the sensitivity of the RX module.
Also, a restructuring here should be considered: I believe it makes more sense to have a section dedicated to the algorithms, then have a section dedicated to the experiments, followed by a section on results and discussion.
As omnidirectional antennas are considered, and due to the NLOS scenario, multipath components cannot be neglected, whereas, detecting an object based on the RSSI is not expected to be accurate. Human detection based on an RSSI change of 1 dBm (Fig. 16a) seems very unreliable. A comparison to literature should be added. Furthermore, a discussion on more realistic scenarios should be added, e.g., what happens if multiple people are moving in the building.
In Figure 6, the legends should be explained, i.e., what do the labels "human x" and "human o" mean?
From Figure 8, it seems that the mean and variance analysis algorithms have the same output, whereas, there are significant differences between the two algorithms in Figure 7.
Typo on line 563: "as shown in Figure 16" should be Figure 13?
The main comment regarding Quality of English Language is conciseness. The paper is very wordy, and some statements are repeated over and over.
For instance: the last sentence of the abstract has the exact same content as the first sentence of the abstract.
Please define abbreviations the first time they are used, e.g., RF, KNN, SI.
Also, some abbreviations are defined multiple times, e.g., BLE.
Please use a hard space between a number and its unit, so that there is no linebreak in between, e.g., at line 59: "433 MHz". In some cases, there is no space between a number and its unit, e.g., at line 122: "100 MHz, 3 GHz".
Also use a hardspace between "Table"/"Figure" and the table/figure number, e.g., at line 409.
In LaTeX, this is done via ~. In Word: https://www.avantixlearning.ca/microsoft-word/how-to-insert-a-nonbreaking-space-in-word/
Please rework some paragraphs. Paragraphs of only a single sentence should be avoided, e.g., at line 554. More info on: https://wts.indiana.edu/writing-guides/paragraphs-and-topic-sentences.html
A full paper is typically organized in Sections, not "Chapters", as mentioned on page 9 (line 371).
It is not necessary that the title of Section 3 ("Overview") is in uppercase as the other section titles are not.
Author Response
I would like to express my sincere gratitude for taking the time to review paper, which was recently submitted to Sensors. Your valuable feedback and insightful comments have been immensely helpful in improving the quality and clarity of my work.

Reviewer 2 Report
This work proposes a human detection solution using 433 MHz frequency bands and presents experimental results for several scenarios. The measurement setup is well-described, and the results look promising. I have some concerns as follows.
1. From lines 393 to 398, the authors mention the laser transmitter and receiver. Why laser is used here instead of RF signal?
2. Is it necessary to use a data-modulated signal ( 8.5 kHz bandwidth) to measure the RSSI? Can a single tone be used instead? Do the transmitter and receiver need to be synchronized?
3. Please explain the setup phase in more details.
Some typos:
1. Punctuation at the end of each equation is missing.
2. Line 453: verification phase should be setup phase.
Author Response

(The authors gave the same response as above.)

Reviewer 3 Report
The work presents an interesting application of wireless RF sensing using transceivers. However. some details should be included to improve the clarity of the work and its presentation:
- A photograph of the measurement setup in Fig. 1 (b) should be included to help visualize the setup and understand how the data was collected.
- A note on the materials used in the building construction, alignment with windows/gaps and any other external factors which influence the propagation should be included.
- The antennas used should be clearly described (using the model number), and preferably, their S11 and gain should be measured to verify their omnidirectionality and efficient operation at the chosen frequency.
- In the figures where Attenuation (dB) is shown on the y-axis, please revise the signs. Negative attenuation (-x dB) means an increase in the gain rather than an increase in the attenuation (-ve -ve signs). Preferably, a different notation should be used not to confuse the reader.
- Table 4 shows that the frequency is around 424 MHz - the abstract and conclusion should be revised to state that the work operates in the 433 MHz band (and not frequency), not to confuse the reader.
- Following on the previous point, the frequency range in the band should be explained in the text.
The work is easy to follow but will benefit from a grammar check using dedicated software.
Author Response

(The authors gave the same response as above.)

Round 2
Reviewer 1 Report
The quality of the manuscript has significantly improved.
Some very minor remarks:
- At line 71, replace “In this paper, we present a new approach for human detection using low-power 433 MHz frequency bands.” by “In this paper, we present a new approach for human detection using multiple low-power transceivers using the 433 MHz frequency band.”
- I appreciate the updated paper structure. It is mich improved compared to the previous version. However, I think you should mention at the end of Section 1 (intro) or 2 (background) that you do the detection based on RSSI signal strength variations after having a setup phase where you get obtain baseline RSSI values (and where the building should be empty).
- What is still not entirely clear is what the efficiency of the considered algorithms is. Based on the the statement that different receiver modules can be used, and that the synchronization is performed using a packet identifier, I guess that the algorithms run offline, i.e., after all measurement data is gathered. A discussion should be added on using this system for real-time human presence detection.
- A comparison of the results to the state-of-the-art should be added. From the text I understand that the system will work better compared to systems at higher frequencies (due to the lower penetration losses). Can the system be quantitatively compared to f.i. the work in [1] or [7] (where also NLOS scenarios are considered)?
The quality of the presentation has improved significantly. Some remaining items are the following.
- Order references according to their first occurrence in the text
- I believe that equations should be referenced as “Equation X” (with capital E), similar to figure references
- There is an added dot in the figure caption of Figure 1
Author Response
Thank you for your time, expertise, and commitment. Your insightful comments, suggestions, and constructive feedback have been immensely helpful in shaping the final version of the manuscript. Your keen attention to detail and expertise in the field have undoubtedly improved the clarity and rigor of my research. Please see the attachment.
